# Building Protein Atomic Models from Cryo-EM Density Maps and Residue Co-Evolution

**DOI:** 10.3390/biom12091290

**Published:** 2022-09-13

**Authors:** Guillaume Bouvier, Benjamin Bardiaux, Riccardo Pellarin, Chiara Rapisarda, Michael Nilges

**Affiliations:** 1Structural Bioinformatics Unit, Institut Pasteur, Université Paris Cité, CNRS UMR 3528, 75015 Paris, France; 2Microbiologie Fondamentale et Pathogènicité, University of Bordeaux, CNRS UMR 5234, 33076 Bordeaux, France; 3Institut Européen de Chimie et Biologie, University of Bordeaux, 33600 Pessac, France

**Keywords:** cryo-EM, co-evolution, model building, minimum spanning tree, type 6 secretion system

## Abstract

Electron cryo-microscopy (cryo-EM) has emerged as a powerful method by which to obtain three-dimensional (3D) structures of macromolecular complexes at atomic or near-atomic resolution. However, de novo building of atomic models from near-atomic resolution (3–5 Å) cryo-EM density maps is a challenging task, in particular because poorly resolved side-chain densities hamper sequence assignment by automatic procedures at a lower resolution. Furthermore, segmentation of EM density maps into individual subunits remains a difficult problem when the structure of the subunits is not known, or when significant conformational rearrangement occurs between the isolated and associated form of the subunits. To tackle these issues, we have developed a graph-based method to thread most of the C-α trace of the protein backbone into the EM density map. The EM density is described as a weighted graph such that the resulting minimum spanning tree encompasses the high-density regions of the map. A pruning algorithm cleans the tree and finds the most probable positions of the C-α atoms, by using side-chain density when available, as a collection of C-α trace fragments. By complementing experimental EM maps with contact predictions from sequence co-evolutionary information, we demonstrate that this approach can correctly segment EM maps into individual subunits and assign amino acid sequences to backbone traces to generate atomic models.

## 1. Introduction

Getting detailed structural information from cryo-electron microscopy (cryo-EM) is a time-consuming process, which involves sample preparation, data collection, particle picking, and image classification, followed by the reconstruction of the three-dimensional (3D) density map. In the early days of single-particle analysis (SPA) EM, this map used to be called “3D structure” [1]. However, the key outcome of a cryo-EM study is an atomic model of a macromolecule or a complex [2]. In the current paper, we will focus on interpreting the final cryo-EM map to generate de novo an atomic model of a protein complex. Therefore, we will ignore the upstream process of obtaining the EM density reconstruction, which will be the input information for the method.

Recent advances in direct electron detectors have led to the structure determination of macromolecular complexes with resolutions as low as 1.2 Å [3,4]. The structural features associated with different resolution ranges are the following:At resolution better than 3.3 Å  the amino acid residue side-chains are visible. This information can be used to unambiguously identify the amino acids and, therefore, to trace the protein sequence into the EM density.From 3.3 to 4.5 Å resolution, side-chains are only partially visible. For this “shadow” range, topological reconstruction of the protein is often tedious unless complementary information is available.For resolution worse than 4.5 Å  only some secondary structures—mainly helices—are visible. The β-strands are no longer separated, rendering the topological reconstruction somewhat impossible.

Whereas the resolution of an EM density map is an overall estimation of its information content, the local resolution can significantly vary [5]. For example, negatively charged residues that are sensitive to radiation damage or flexible regions have a local resolution that is sensibly lower than the global one. As a consequence, parts of the map may be difficult to interpret, even if the global resolution is sufficient.

Several tools have been proposed to reconstruct protein models from EM density maps. The most common approach is to use structures of the individual components of a complex obtained in isolation by NMR, X-ray crystallography, or comparative modeling. The structures are docked as rigid bodies into the EM map [6]. These structures can then be further refined, allowing for local flexibility to maximize the correlation with the EM map. This method breaks down when no 3D structures are known. For the highest resolution cases, tools developed for X-ray crystallography are usually applied to cryo-EM [7,8], such as Coot [9]. Even though automation tools are provided within Coot, the process is largely manual, time-consuming, and prone to subjectivity. Furthermore, the sequence assignment is mainly based on side-chain densities to identify the amino acids, which can be impossible in certain parts of the protein.

Other tools have also been developed specifically for EM data [10,11]. One of these methods is based on Rosetta [12] and makes use of the predicted backbone conformation to assign the protein sequence in maps where side-chain density is ambiguous. Recent advances in machine learning, and especially in deep learning (DL), has led to the development of new approaches using this technology. Those approaches often used 3D-convolutional neural networks to find the location of Cα atoms of the protein backbone. The tracing of the chain is usually made by using a graph theory algorithm, such as the traveling salesman problem (TSP) formalism. Recent state-of-the-art approaches include DeepTracer [13], EMBuild [14], or CR-I-TASSER [15], which predict main-chain probability maps for the positioning of Cα atoms in the density. These DL methods are usually trained on large sets of experimental and simulated EM maps along with the corresponding known PDB models. The procedure described in the current paper is not a direct competitor of these new DL approaches, but is proposed as a new pipeline that can be combined with existing methods to help the tracing for difficult cases when those approaches are not entirely successful to produce a correct tracing of the protein backbone. Therefore, instead of benchmarking our method directly with those different approaches, we rather chose to present a usage test case where it has been particularly successful and helped to solve a previously unsolved structure such as the wedge complex of the bacterial type 6 secretion system (T6SS) baseplate [16].

One of the first challenges of interpreting de novo the EM map of a multi-component complex is the map segmentation and protein identification. Map segmentation is a method that divides the map into submaps, each of which corresponds to a subset of subunits in the complex. Other tools exist to find densities corresponding to the asymmetric unit [17], but if the asymmetric unit is composed of distinct protein chains, to further segment the map accurately remains challenging.

The goal of our methodology is to simultaneously segment the EM density map of the asymmetric unit (ASU) and build an atomic model of all individual protein chains in the ASU. In the rest of the manuscript, we will thus call “segmentation” the process of identifying map densities corresponding to each individual subunit in the ASU. To do so, we use contacts between amino acids predicted from both evolutionary couplings (EC) and sequence conservation. Usage of EC data to help structure building has already been used in other approaches. However, those data have been mainly used in manual structure building, for example to find the relative orientations of alpha-helices in a protein structure [18]. In the current approach we aim to implement the EC data as an integral part of the model-building process.

## 2. Materials and Methods

An EM density map is stored as a cubic grid, where each grid point encodes the molecular density. As an input, the algorithm described below requires prior identification of the map corresponding to the asymmetric unit (ASU). The map of the test case is ASU of EMD-2513, which is composed of 3 protein chains, and has been obtained by using UCSF Chimera’s “zone tool” [19] and results in a grid of 80×74×76 grid points with a spacing of 1.32 Å.

### 2.1. Tree Representation of an EM Map and Minimum Spanning Tree

The first step of our method is to describe the EM map as a weighted graph G(V,E). The vertices v∈V are the grid points of the EM map, and the edges e∈E are the 26 direct neighbors in the grid. The weight $w_{i,j}$ of the edge connecting vertices $v_{i}$ and $v_{j}$ is defined as the inverse of the product of the map density of the corresponding grid points noted as $d_{i}$ and $d_{j}$, respectively, $w_{i,j}={\left(d_{i}\times d_{j}\right)}^{-1}$.

If we assume that the protein backbone is a high-density feature of the EM map, then determining the chain trace should correspond to finding the path of minimal weight in such a graph. Therefore, a useful way to simplify the graph is to build the minimum spanning tree (MSTree) of the graph. By definition, the MSTree is a subset of edges of the graph that connects all the vertices without any cycles and with the minimal total weight. Therefore, the MSTree should encompass the correct backbone threading in the EM map. We use the Kruskal algorithm [20] to build this tree from a graph *G* built on the EM density.

In a third step, the tree is pruned by removing portions of the graph until the maximal degree of the graph, i.e., the maximal number of edges per vertex, is three. This simple pruning rule allows us to obtain a potential candidate for the threading of the Cα trace through the density because Cα atoms are covalently bonded to three non-hydrogen atoms at most. We note that the MSTree is similar to the result of the skeletonization procedure often used in de novo model building from density maps to guide manual threading of the Cα trace [9].

The final goal of the graph algorithm is to obtain the longest fragments describing the best possible partial Cα threading of the protein backbone. Therefore, as the spacing between the nodes of the graph is still equal to the spacing of the input EM density map (1.32 Å in our case), one must place Cα atoms as accurately as possible. Starting from remaining vertices of degree three, which are likely Cα atoms with visible density for the side-chains (or at least Cβ atoms), vertices coordinates are refined to get as many vertices separated by 3.8Å as possible—i.e., the average distance between two consecutive Cα carbons (Figure 1a).

Because the resulting graph can still contain forks, we developed an additional pruning procedure to suppress remaining forks and to get unbranched fragments that approximate the backbone threading in the EM density map (Figure 1).

### 2.2. Fragment Assembly and Map Segmentation

The fragments obtained from the procedure above need to be assembled to build the protein structure, and the sequence of the chain (or several chains) has to be assigned to the detected Cα positions. For this purpose, we use the predicted contact map of the chain. This part of the procedure is based on the map_align algorithm developed by Ovchinnikov et al. [21]. The map_align algorithm takes two contact maps and returns an alignment that tends to maximize the number of overlapping contacts while minimizing the number of gaps. Along with the alignment, map_align gives a score for the quality of the alignment.

However, the original algorithm was designed to match predicted contacts with contact patterns of known protein structures and not for de novo building of protein chains from fragments. In order to apply the map alignment tool to this purpose, we developed an algorithm that iteratively concatenates the fragments built to maximize the alignment score. For an arbitrarily chosen chain of the complex, the algorithm finds the fragment and its direction that maximizes the map_align score with the predicted contact map. For this current fragment, the algorithm considers all its neighbors and calculates their tentative map_align scores. The newly calculated tentative map_align scores are compared to the current ones, and the new current fragment is chosen as the fragment with the highest score, and so on. The algorithm stops after a chosen number of iterations or when all combinations have been tested. Dynamic programming allowed us to find the subset of fragments and their associated directions by maximizing the overlap with the predicted contact map for each chain. In this process, intermediate fragment merging alignment scores are cached to avoid recomputing them back for further chain elongation. A pseudocode for the merging procedure of the fragments is proposed in Appendix A.

Once the first chain has been built, the full set of fragments is reused to get the most probable second chain, and so on for the remaining chains. Then the chains are sorted by alignment scores. The fragments used to build the chain with the highest score are removed from the pool of fragments, and this chain is considered as assigned. Then, the second-highest scoring chain is built in the same way, but from the new subset of fragments, preventing overlap of the newly built chain with the previous one, and reducing the combinatorial complexity of the problem simultaneously. This is repeated for each chain until the last chain is built. In this way, the algorithm proposes a segmentation of the EM map based on the predicted contacts.

To compute the contact maps derived from the fragments, we use a distance cutoff of 8 Å between Cα atoms. Furthermore, the RaptorX contact prediction server provides a score, given as a probability, along with each contact identified. For the current test case, we used a probability threshold of 0.5 to derive the predicted contact maps. Such a threshold is a commonly accepted value when generating binary contacts classification from contact prediction with an associated probability [22].

Lastly, the final contact map alignments allow registering the sequence of the subunits onto the Cα traces as it provides the correspondence between a bead position and the corresponding amino acid position in the sequence used to predict the contacts.

### 2.3. Building the Full Atomic Model

In the final step, we use Modeller [23] to obtain a full-atom model of the structure. The obtained model is further refined in the EM map by using the real_space_refine tool [24] from the Phenix suite. During the refinement, secondary structures are optimized by using an optional feature in Phenix that annotates secondary structure based on Cα positions.

## 3. Results

### 3.1. Method Principle

The problem of tracing a chain in the EM density map of a multi-component complex can be divided into several tasks: (i) the map is segmented into submaps of the individual components; (ii) probable locations of amino acid residues are identified in the map; (iii) these locations are assigned to the primary sequences of the components; and (iv) full-atom models of the protein are constructed.

The method described in more detail below has four major stages:(1)The cryo-EM map is initially represented as a weighted graph and simplified into a minimum spanning tree (Figure 2b) (MST). This tree is pruned until the maximal degree of the graph (i.e., number of edges of the vertex with the greatest number of edges incident to it) is three.(2)The tree is further pruned to remove forks in the tree, and to produce potential fragments of the polypeptide chain. At this stage, the fragments have an arbitrary N-to-C terminal orientation.(3)The fragments are concatenated by making use of evolutionary information in the form of predicted inter-residue contacts. We used the Raptorx ContactMap server [25] to predict the contacts. The direction of each fragment is determined to maximize the agreement with the predicted contacts. During this stage, the map is segmented automatically, as a consequence of the fragment concatenation procedure. No prior segmentation of the map to individual components is necessary.(4)In the final stage, a full-atom model is constructed and refined against the map by standard approaches, Modeller [23] and the real_space_refine tool [24] from the Phenix software suite, respectively.

### 3.2. Application to the F420-Reducing Hydrogenase Heterotrimer

As a test case, we applied our method to the cryo-EM map of the F420-reducing hydrogenase (EMD-2513) determined at 3.36 Å resolution [26]. The map has regions of high and low resolution, correspondingly 3.3 Å and 6.6 Å according to ResMap [5]. The deposited model (PDB ID 4CI0), composed of 3 chains of 386, 275, and 281 amino acids, respectively, was obtained by a real-space refinement of a former model derived from a previous lower resolution map [27], which in turn had been built by the manual fitting of homologous protein structures and subsequent refinement. Regions for which no homolog was available had been manually built by using Coot [9]. In stark contrast, the overall process of our method is independent of any prior structural knowledge of a homologous protein. We note that in X-ray crystallography, much higher resolution is required for automated model building [28].

The inputs of the algorithm are the map of the asymmetric unit consisting of a heterotrimeric protein complex (Figure 2a), the primary sequences of the three protein subunits, and the predicted contacts for each subunit. RaptorX provides a score, given as a probability, for each predicted contact. We used a cutoff of 0.5 on the prediction score to discard irrelevant contacts, ending up with 658, 343, and 563 predicted contacts for chains A, B, and C, respectively. The average number of contacts per residue is consequently 1.7, 1.2, and 2.0 for the 3 chains, respectively.

For our test case, the algorithm found 18 fragments (Figure 2c) with an average accuracy of 1.2 Å, computed as C-α RMSD to the reference structure, 4CI0.

The generated fragments fully cover the three chains of the reference structure 4CI0 (Figure 3). In general, the fragments tend to correspond to clusters of secondary structure elements, and the ends of the fragments lie at the edges of secondary structure elements.

As mentioned, the identified fragments do not contain any information on the directionality of the polypeptidic chain (N-to-C or C-to-N). Hence, both directions of the fragments have to be tested to assign the protein sequence, thus considerably increasing the combinatorial complexity of the problem, as twice the number of fragments have to be considered. Chain direction inversions are clearly apparent as “crossed” features in chain A and C (Figure 3).

The resulting identified chains are shown in Figure 2d. In the example, the proposed segmentation fits extremely well with the chains of the reference structure (PDB 4CI0). In contrast to other tools developed for de novo protein structure determination from EM maps, which consider segmented maps for individual chains as a prerequisite [2,12], as mentioned earlier, the current approach resolves map segmentation and tracing of the chain simultaneously.

We compare the final backbone contact maps with the predicted contact maps in Figure 4. The overlap score is defined as the fraction of predicted contacts also observed in the model. To get a local overlap, we define a window of 11 consecutive residues and slide it along the sequence. We compute the overlap score of the 11-mer with the target structure for each residue position (Figure 4). In order to assess the accuracy of the sequence assignment based on the predicted contact map, we color-coded the computed structure according to the overlap score (Figure 4d). Even if the average overlap score with the predicted contacts is rather low, it still allows us to identify regions with poor overlap and potentially incorrect sequence assignment. The overlap scores is much higher when compared with the contacts observed in the true solution, i.e., the reference PDB structure, but such a comparison is possible in a real-case scenario when the true solution is not known (Appendix A).

Next, we compared the obtained model to the original reference structure 4CI0 (Figure 5). Deviations of Cα positions from the reference are plotted in Figure 5a,b and c for chains A, B, and C, respectively. Chains A and C are accurately placed with an average deviation of 2.75 Å and 2.31 Å  respectively. Globally, regions with higher deviations are contiguously found around residues 250 for chain A—noted with symbol † in Figure 5a—and for the C-terminus of chain C only—noted with an asterisk in Figure 5c—and are delimited well in the structure (Figure 5d). These regions correspond to the regions where the predicted contacts are sparser than for the rest of the protein, highlighted above with symbols † and *. Chain B is less accurately placed with an average deviation of 4.17 Å. The deviation is essentially due to residues 150 to 200—noted § in Figure 5b. This deviation is detected by the low overlap score obtained for that region (§ marker in Figure 4b), and explained by the very sparse predicted contacts, combined with the local low resolution of the cryo-EM map.

Additionally, we computed the local correlation per amino acid residue to assess the fit to the cryo-EM data of the refined reconstructed model (Figure 6). For this, we used the segment Manders’ overlap coefficient (SMOC) as described in [29] with a sliding window of 11 residues, similarly to the calculation of the overlap score. Qualitatively, the protein regions that diverged in Cα positions from the reference structure are also poorly fitted in the density, especially for the region noted § for chain B (Figure 6b). Therefore, the local correlation of the obtained model with the EM density map can be used to identify regions to carefully analyze and manually refine if necessary when no structural data is available. An incorrect register of the sequence can explain those results due to poorly predicted contacts for this region or to a poor local resolution of the density map.

### 3.3. Application to the T6SS Baseplate with Manual Tracing

The method was also adapted for the use with previously obtained backbone trace, e.g., from manual tracing. We applied this modified methodology to determine the structure of the wedge complex of the bacterial Type 6 secretion system (T6SS) baseplate [16]. The wedge complex is mainly composed of three proteins: TssF, TssG, and TssK. The cryo-EM density map (EMD-0008) is organized into three lobes, with distinct resolutions. Two of the lobes were occupied by two TssK trimers. In these regions, the map resolution and structural prior information were insufficient to build accurate atomistic models. In contrast, the resolution of the lobe corresponding to TssG and TssF was between 4.3 Å and 8 Å. Because only weak homology information was available for TssG and TssF, the structure of the proteins was determined de novo by using a semi-automatic iterative pipeline, fully described in [16].

In this application, we did not employ the automatic backbone tracing (steps 1. and 2. in Section 3.1) because the resolution was too low for an automatic tracing procedure. The TssFG map was segmented by using Segger [30], which identified densities corresponding to TssG and two TssF subunits arranged in a pseudo-C2 symmetry. The C-α backbone structure was traced manually based on the densities of the three segments by using Coot. The tentative sequence registering was obtained by aligning the bulky amino acids and using secondary structure prediction. However, the manually traced backbone was highly inconsistent with the predicted contacts (Figure 7a,f). To resolve these inconsistencies, we iteratively applied two steps: first, we fragmented the backbone trace based on the alignment of the model contact map onto the predicted contact map. The alignment identified gaps with more than four residues, which we used to guide the fragmentation. Second, we concatenated the fragments following stage 3 in Section 3.1. For TssF, six iterations of fragmentation and concatenation were required to obtain a model fulfilling most of the predicted contacts (Figure 7b,g).

With respect to the topology of the manually traced structure (Figure 7d), the method reoriented three beta-strands, two helices, and changed the connectivity of four pairs of secondary structure elements, localized in the center of the sequence (red secondary structure elements in Figure 7e). These changes had a huge impact on the topology, considering that two-thirds of the sequence was offset by about 100 residues (dark gray secondary structure elements in Figure 7e).

We were able to build an atomistic model of TssF and TssG (not shown) components of the complex, in which most of the sequence of the proteins could be assigned to the cryo-EM density and secondary structure elements could be identified (Figure 7c). This same structure was later confirmed by other authors by using higher resolution cryo-EM maps [31], with Cα RMSD of 3.6Å and 1.8Å for TssF and TssG, respectively, confirming the accuracy of the method.

### 3.4. Additional Challenging Targets

To assess the limits of our algorithm, we applied it to two additional targets: (i) the E. Coli beta-galactosidase (PDB 3J7H) at 3.2 Å resolution [32] and (ii) and the human gamma-secretase complex (PDB 5A63) at 3.4 Å resolution [33] (Appendix A). Beta-galactosidase is a homo-tetramer with D2 symmetry. We used a submap from EMD-5995 corresponding to a single protomer where 1022 residues have been modelled in the reference PDB structure. Our MSTree algorithm resulted in high Cα accuracy fragments, covering 94% of the reference PDB trace (Appendix A). However, the algorithm failed to correctly assign the sequence on the obtain Cα. This result could be attributed to long breaks between some fragments, wherein the distance between fragment ends is larger than the chosen threshold (11 Å), preventing them from being correctly joined during the merging procedure. The breaks correspond to regions with low density in the map at the chosen level (Appendix A). When the fragment merging and sequence assignment procedure is applied to ideal fragments (fragments of random length with coordinates of the Cα atoms from the reference PDB structure), an almost perfect reconstruction of the model is obtained with the predicted contacts (Appendix A).

The gamma-secretase is a 4-chain complex, so the full density map (EMD-3061) without prior segmentation was used as input. Interestingly, it was possible to reconstruct models with medium accuracy only for chains A and C (Cα RMSD to the reference PDB structure of 5.7 Å and 4.7 Å, respectively). As for the beta-galactosidase, breaks between fragments along with forked fragments creating shortcuts in the tracing prevented the correct sequence assignments of chains B and D (Appendix A). For chain B, 50% of the density was correctly detected but with incorrect sequence register, and part of chain D was erroneously assigned to the rest of the density corresponding to chain B (Appendix A). This result can be also partially explained by the lower number of predicted contacts for chains B, C, and D with <0.7 contacts per residue (Appendix A). However, when ideal fragments from the reference PDB are used as input, generated models for chains A and C, which were well modeled with fragments from the MSTree approach, yielded RMSD lower than 1 Å. The RMSD for chains B and D were slightly higher (1.3 Åand 2.4 Å, respectively). It is noteworthy that the coverage (percentage of modeled residues) for chains B, C, and D, even with ideal fragments, is below 85%, confirming that the sparsity of the predicted contacts hampers correct merging of all fragments and threading of the full-length sequence (Appendix A).

These two targets represent challenging systems with either a long single-chain subunit (3J7H) or multiple chains with sparse, predicted contacts (5A63). Despite producing majorly accurate Cα fragments, missing fragments, forked fragments, or insufficient information in the predicted contacts prevent accurate reconstruction of the full molecule. Nevertheless, the use of evolutionary contacts still proves efficient in accurately reconstructing atomic models from EM maps when ideal fragments are provided.

## 4. Discussion

In the first test case, our method resulted in a model with an average accuracy of 2.4 Å computed as the C-alpha RMSD to the deposited structure. The EM density map was segmented in the process of sequence assignment of individual protein chains, with only sparse and noisy contacts from residue co-evolution, and, remarkably, without the use of 3D structures of homologous proteins.

A recent strategy, MAINMAST [10], also makes use of the MSTree in the early stages of the modeling. However, the tree is refined by using a Tabu-search algorithm, and the algorithm is applied to already segmented maps, due to the complexity of the combinatorial problem with larger systems containing multiple subunits. Furthermore, the MAINMAST approach employs a threading approach to register the target sequence on the traced chain by evaluating the fit of the amino acid sequence of the protein to a path in a tree. Our algorithm uses a different approach as the sequence is registered by using an alignment procedure between the contact map of the protein model being built with the contact map predicted by using residue co-evolution.

Obviously, in some cases, manual intervention will eventually be necessary. The overlap score (Figure 4) helps to identify regions of the protein that might be problematic. The test case illustrates how the overlap function correlates with divergence from the reference structure 4CI0 (Figure 5). Hence, this score provides an objective metric with which to identify troublesome regions where local modification of the 3D structure might be necessary, or place fragments and assign their sequence by hand to maximize the overlap with the map. Furthermore, when the minimum spanning tree and pruning procedures lead to incorrect topology, the overlap score should pinpoint the misleading regions, thus allowing one to fix the error if possible.

The fragment placing and merging procedures, and the sequence assignment are two independent processes in the workflow. Therefore, the method is very flexible, as any fragment building method can be coupled with the merging procedure based on the predicted contact maps. In particular, manual tracing of fragments can be easily coupled with our sequence assignment method in cases where the map is of insufficient resolution or quality to place atoms or fragments automatically. In this spirit, we have used the fragment merging and sequence assignment functionalities to solve the unknown structure of the TssF protein, a subunit of the T6SS baseplate, by using a 4.7 Å resolution map in combination with manual tracing [16].

For homo-multimers, the ASU corresponds to the protomer structure, so our approach would model it as a single chain. Of course, contact prediction may contain contacts corresponding to inter-protomer interactions that would not be fulfilled during our fragment merging and contact map alignment procedure, and could in principle impair its efficacy. In the current implementation, inter-protein contacts are not yet taken into account, but the method can be readily adapted. Alternatively, predicted inter-subunit contacts can still be used with the existing implementation to validate the final model. The number of subunits and chain length (i.e., number of residues) might also limit the efficacy of our approach because (i) the merging procedure relies on a full pairwise estimation of the fragments pairing and for which the complexity increases along with the number of fragments, and (ii) each chain is processed sequentially. As demonstrated with the two additional examples, missing fragments for low-density regions also hamper the correct registering of the sequence with the predicted contacts.

The major limitation of the method is at present the collection of reliable contacts from residue co-evolution. Indeed, a sufficient number of homologous sequences is required to predict contacts with an acceptable degree of reliability [34]. This limitation will likely be of less importance in the future with the availability of contact prediction methods that require fewer sequences [35].

## Figures and Tables

**Figure 1 biomolecules-12-01290-f001:**
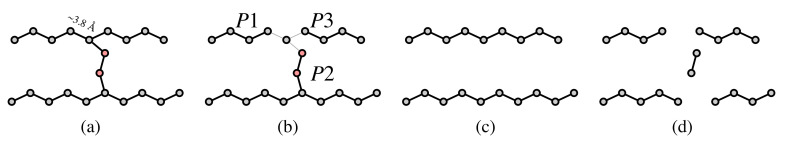
Fork pruning procedure. (**a**) Regularization of the nodes of the MSTree to be spaced by 3.8 Å. The anchoring Cα defining the reference position to regularize the positions are defined by nodes of degree 3. These nodes are involved in forks (in red) that must be pruned to recover possible backbone threading. (**b**) When a fork is detected, the edges of the fork are temporarily removed, which creates 3 possible path graphs (P1, P2 and P3). (**c**) The shortest path graph (P2) is removed to solve the fork. However, a pruning threshold ($T_{p}$) is defined to avoid the removal of large fragments. If the length of the path graph is lower than this threshold, the path graph is removed, and the fork is solved. (**d**) In the case in which the length of the shortest path graph is higher than $T_{p}$, the fork nodes are simply deleted and more fragments are generated.

**Figure 2 biomolecules-12-01290-f002:**
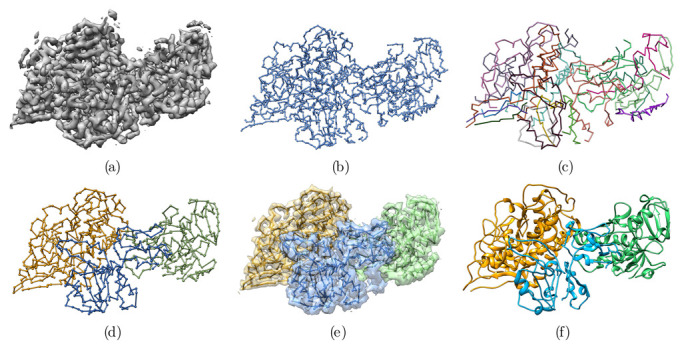
Overview of the automatic reconstruction of an atomic protein structure from an EM density map. (**a**) Original EM density map used as input of the algorithm. (**b**) Minimum spanning tree (MSTree) derived from the EM density map. (**c**) Fragments resulting from the fork pruning procedure of the MSTree. (**d**) Chains built from the fragments. Fragments were merged in a way to maximize the overlap with predicted contacts. Chains A, B, and C were detected automatically without any prior segmentation of the EM density map. (**e**) Segmented map built from the threading of the 3 chains depicted in (**d**). (**f**) Full-atom model derived from the Cα tracing in (**d**) with Modeller and the real_space_refine tool from the Phenix software suite.

**Figure 3 biomolecules-12-01290-f003:**
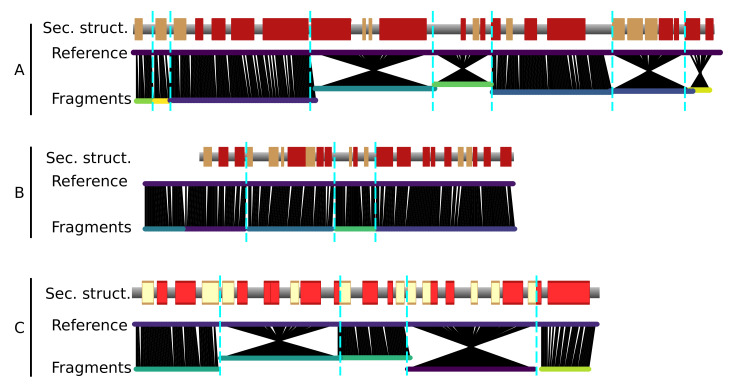
Fragments resulting from the map threading (bottom lines) and aligned on the reference model 4CI0 (upper lines) for the 3 chains (**A**–**C**). The red and yellow boxes indicate α-helices and β-strands, respectively, in the reference model. The black lines indicate the nearest Cα atom in space from the fragment to the reference model. Crossed line bundles indicate fragments that have been reversed with respect to the originally predicted direction of the graph.

**Figure 4 biomolecules-12-01290-f004:**
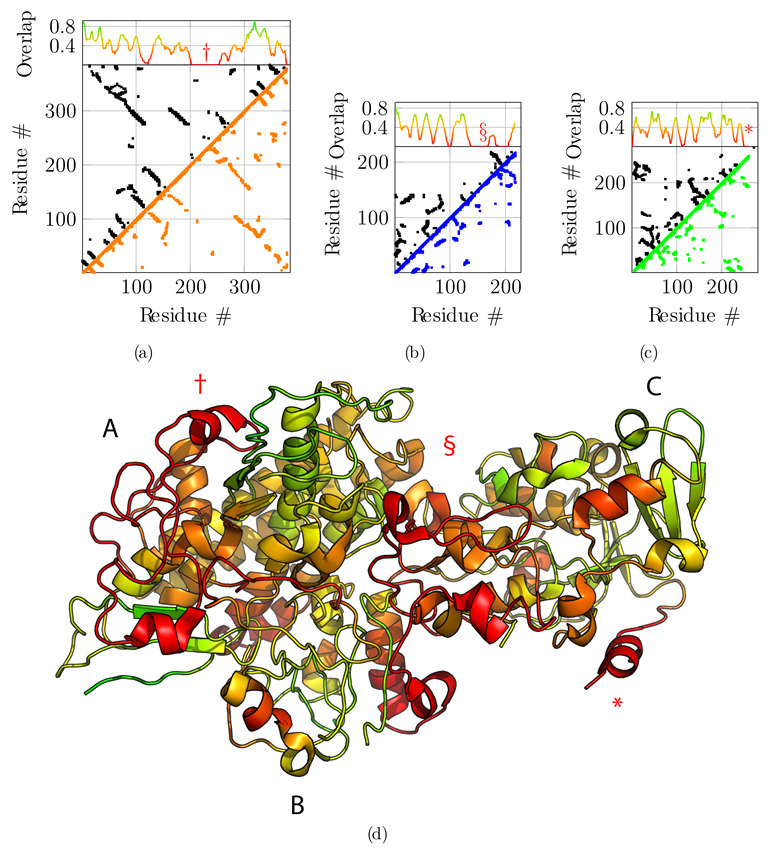
Overlap between predicted contact maps and contact map of the final model. Predicted contact maps (upper diagonals, in black) with a probability cutoff of 0.5, are compared with the final contact maps of chain A, B, and C ((**a**), (**b**) and (**c**), respectively) resulting from fragment merging (lower diagonal in orange, blue, and green, respectively) with a distance cutoff of 8 ÅṪhe upper projection gives the overlap ratio between the contact map of the model and the predicted contact map. An overlap of 1 means that all predicted contacts are satisfied by the model. The overlap is computed for sliding fragments of 11 residues along the sequence. (**d**) Projection of the contact map overlap on the generated model, colored from red (overlap 0) to green (overlap 1). Three regions, highlighted by symbols †, § and * for chains A, B, and C, respectively, having very sparse predicted contacts display very low overlap scores, indicating that the sequence assignment for these regions on the final model is potentially incorrect.

**Figure 5 biomolecules-12-01290-f005:**
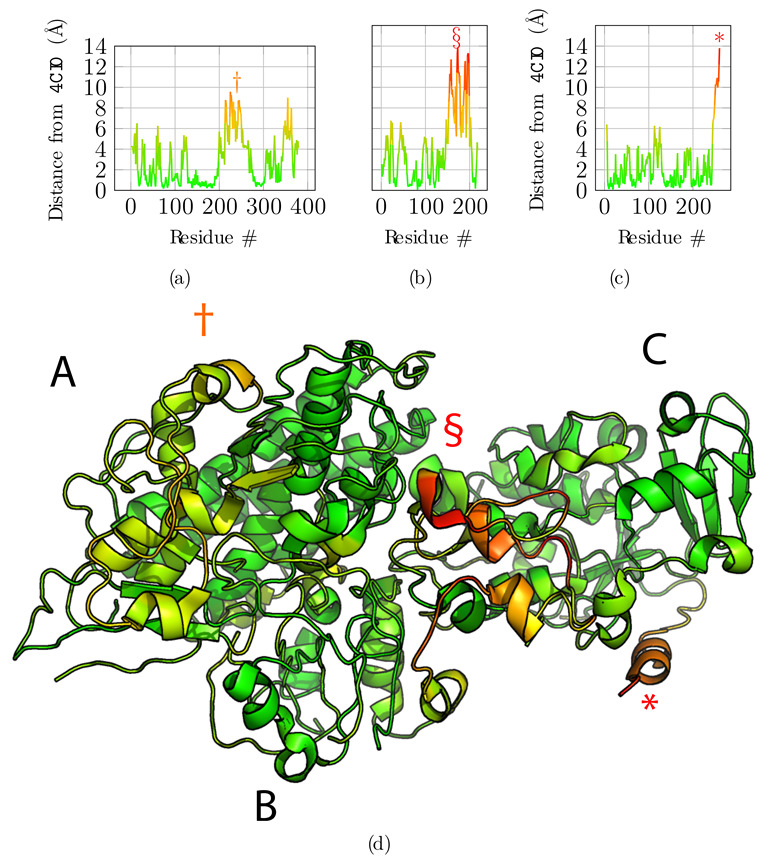
Structural alignment and comparison with the reference structure 4CI0. (**a**–**c**) Deviations of the final model for each of the Cα atoms from the 4CI0 structure aligned on the model, for chains A, B, and C, respectively. (**d**) Final model with Cα deviations projected onto the structure using the same color map as for graphs (**a**–**c**). Symbols †, §  and * highlight regions in the structure with high deviations from the reference structure for chains A, B, and C, respectively.

**Figure 6 biomolecules-12-01290-f006:**
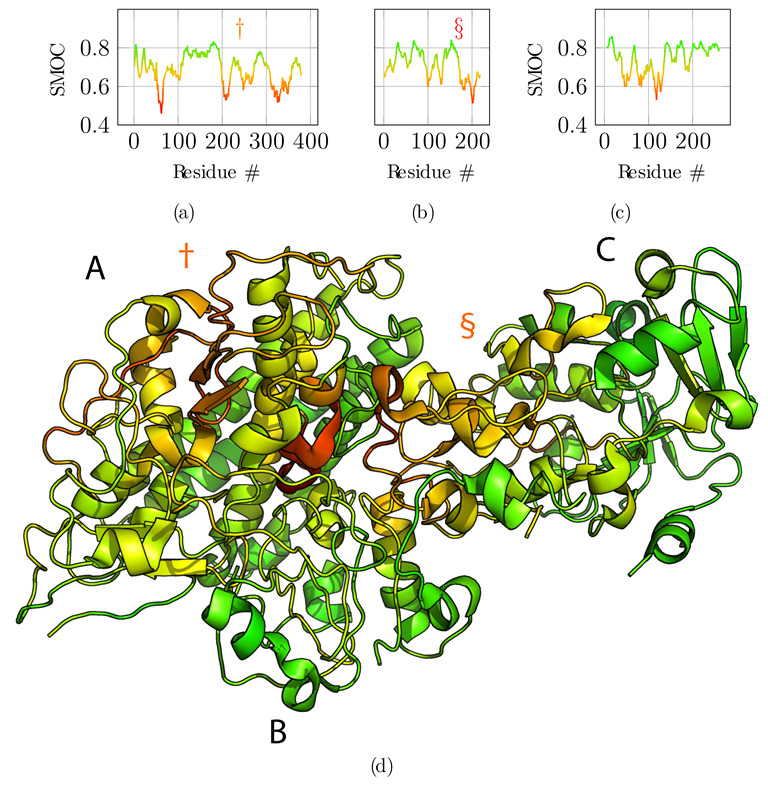
Segment Manders’ overlap coefficient (SMOC) of the refined model against the EM density map EMD-2513 used as input for the atomic scale reconstruction method. (**a**–**c**) SMOC profiles along the protein sequence of chain A, B, and C, respectively. (**d**) The corresponding SMOC values are projected on the obtained full-atom model. The color scale highlights poorly fitted regions in red. Symbols † and § indicate two of the tree regions previously identified as having high deviation from the reference structure.

**Figure 7 biomolecules-12-01290-f007:**
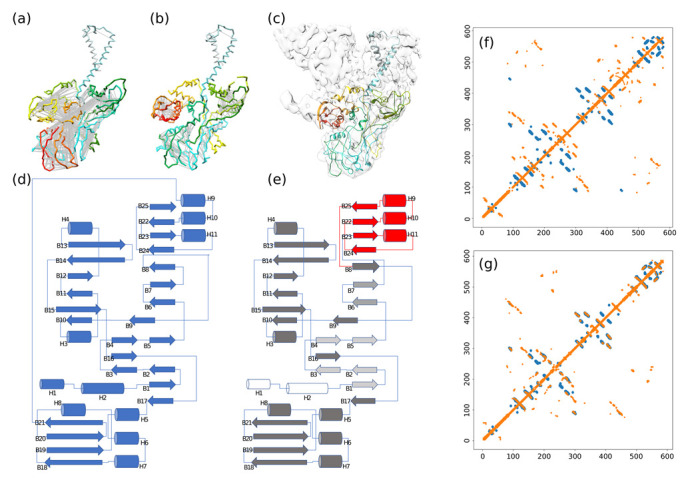
Modified methodology to determine the structure of the wedge complex of the bacterial type 6 secretion system (T6SS) baseplate. (**a**) Initial model manually built in the EM density map. (**b**) Corrected model using predicted contacts. Gray lines correspond to contacts predicted by RaptorX, mostly unsatisfied in the initial model. The rainbow color of the backbone reports the residue index, where N-terminal residues are blue, and C-terminal residues are red. (**c**) Final atomic model docked in the EM map to the wedge complex. (**d**) Topology of the initial model depicted in (**a**). (**e**) Topology of the corrected model depicted in (**b**). The red part highlights the major topology changes made by the method (fragmentation/concatenation). The gray scale encodes the sequence offset introduced by the method: white secondary structure elements have no offset with respect to corresponding secondary structure of the initial topology, light gray elements have an offset of about 10 residues, dark gray have an offset of about 100 residues. (**f**) Contacts of the initial model (orange) aligned on the predicted contacts (blue). (**g**) Contacts of the final model (orange) aligned on the predicted contacts (blue).

## Data Availability

Source code, usage guidelines and example data are available at https://github.com/bougui505/EMEC, accessed on 1 September 2022.

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
