# Peer review of "Building Protein Atomic Models from Cryo-EM Density Maps and Residue Co-Evolution"

_biomolecules, 2022, doi:10.3390/biom12091290_

Round 1
Reviewer 1 Report
The manuscript by Bouvier et al. describes a method for building de novo atomic models of proteins in medium resolution maps. The authors describe their approach based on two model systems, the F420 reducing hydrogenase at 3.4Å resolution and the T6SS baseplate at an average lower resolution. The manuscript describes the overall approach in sufficient detail, there are however three major points that need to be addressed before publication.
1. The authors emphasize multiple times that identifying individual (asymmetric) subunits in an EM map is a critical step. Both of their examples do however rely on prior information (the model of the ASU for the hydrogenase and manual segmentation for the T6SS baseplate). How does the proposed algorithm work if there is no prior knowledge of the contents of the ASU or will it always require manual segmentation? If yes, this should be clarified as sentences like the one on line 77 imply some degree of automated segmentation. If this statement only applies to the contents of an asymmetric unit, this needs to be clearly stated.
2. I am unable to find any reference to a practical implementation of the proposed spanning tree approach described in section 2.1. If this manuscript wants to have a positive impact on the problem of building de novo models in cryo-EM maps it should not be left to the reader to write their own implementation of the proposed approach.
3. In general, the manuscript is a bit sparse in describing the necessary details to repeat the results or attempt to apply them to a new problem. E.g. how exactly were the fragments merged to provide a maximum overlap with predicted contacts? The manuscript mentions dynamic programming, but that is not sufficient to reproduce the steps.
Addressing these concerns will make the contents of the manuscript more approachable to a general audience and help to apply the described principles to other molecules and complexes.
In addition to the point mentioned above, these minor points should be addressed as well:
Line 28: This information is outdated. The highest resolution achieved by SPA is ~1.2Å
Line 62: should be corrected to: “One state-of-the-art approach is DeepTracer.”
Line 64/65: combined with
Line 87: …of EMD-2513
Line 153: What other values were tested, and would it be possible to provide a reasonable range for an interested reader that wants to test this on their own data?
Figure 2 caption: There are two (( in front of e
Author Response
Comments:
The manuscript by Bouvier et al. describes a method for building de novo atomic models of proteins in medium resolution maps. The authors describe their approach based on two model systems, the F420 reducing hydrogenase at 3.4Å resolution and the T6SS baseplate at an average lower resolution. The manuscript describes the overall approach in sufficient detail, there are however three major points that need to be addressed before publication.
- The authors emphasize multiple times that identifying individual (asymmetric) subunits in an EM map is a critical step. Both of their examples do however rely on prior information (the model of the ASU for the hydrogenase and manual segmentation for the T6SS baseplate). How does the proposed algorithm work if there is no prior knowledge of the contents of the ASU or will it always require manual segmentation? If yes, this should be clarified as sentences like the one on line 77 imply some degree of automated segmentation. If this statement only applies to the contents of an asymmetric unit, this needs to be clearly stated.
The method relies on extracts of the density map corresponding to the asymmetric unit (ASU). Tools exist to detect the density of the ASU in cryo-EM density maps (such as the Phenix tool segment_and_split_map, [doi: 10.1016/j.jsb.2018.07.016]) or it can be obtained by visual analysis of the map. For complexes with no symmetry, the ASU would be the entire density. We specified this requirement in the Material and methods
“As an input, the algorithm described below requires prior identification of the map corresponding to the asymmetric unit (ASU).”
In the text, we use the term segmentation as the “identification of densities corresponding to each individual subunit in the ASU”. We understand that it could confuse the reviewer and the readers, so made this more clear in the introduction:
“In the rest of the manuscript, we will thus call "segmentation" the process of identifying map densities corresponding to each individual subunit in the ASU.”
- I am unable to find any reference to a practical implementation of the proposed spanning tree approach described in section 2.1. If this manuscript wants to have a positive impact on the problem of building de novo models in cryo-EM maps it should not be left to the reader to write their own implementation of the proposed approach.
We have added the link to the Github repository of our algorithm (https://github.com/bougui505/EMEC) in the “Data availability” section. The repository contains the source code, usage guidelines and example data.
- In general, the manuscript is a bit sparse in describing the necessary details to repeat the results or attempt to apply them to a new problem. E.g. how exactly were the fragments merged to provide a maximum overlap with predicted contacts? The manuscript mentions dynamic programming, but that is not sufficient to reproduce the steps.
First, the source code is now released along (see “Data availability” section) with example data. Furthermore, we added a sentence to clarify the dynamic programming algorithm and added a pseudocode in supplementary information (Algorithm S1):
“In this process, intermediate fragment merging alignment scores are cached to avoid recomputing them back for further chain elongation. A pseudocode for the merging procedure of the fragments is proposed in Algorithm S1.”
Addressing these concerns will make the contents of the manuscript more approachable to a general audience and help to apply the described principles to other molecules and complexes.
In addition to the point mentioned above, these minor points should be addressed as well:
Line 28: This information is outdated. The highest resolution achieved by SPA is ~1.2Å
We have updated the reference to Nakane et al. (2020) for the 1.22 Å resolution reconstruction of mouse apoferritin.
Line 62: should be corrected to: “One state-of-the-art approach is DeepTracer.”
Corrected.
Line 64/65: combined with
Corrected.
Line 87: …of EMD-2513
Corrected.
Line 153: What other values were tested, and would it be possible to provide a reasonable range for an interested reader that wants to test this on their own data?
We use the 0.5 probability threshold since it is a commonly accepted value when generating binary contacts classification from contact prediction with an associated probability (as used by CASP in their evaluation of such prediction methods (Schaarschmidt, J, Monastyrskyy, B, Kryshtafovych, A, Bonvin, AMJJ. Assessment of contact predictions in CASP12: Co-evolution and deep learning coming of age. Proteins. 2018; 86: 51– 66. https://doi.org/10.1002/prot.25407). We have added the following sentence to the Methods section.
“Such a threshold is a commonly accepted value when generating binary contacts classification from contact prediction with an associated probability [22].”
Figure 2 caption: There are two (( in front of e
Corrected.
Reviewer 2 Report
The authors have developed the method for generating an atomic model of a protein complex, based on the cryo-EM map. Their method resolves simultaneously a segmentation problem of the EM map and a construction problem of atomic models for all chains in a protein complex. To this end, they try to use predicted contacts between amino acids, derived from both evolutionary couplings and sequence conservation. Their algorithm proposes a segmentation of the EM map based on the predicted contacts. They show the applicability of their method for the F420-reducing hydrogenase heterotrimer and show the applicability of their modified method, which utilize the obtained backbone trace, for the wedge complex of the bacterial T6SS baseplate.
Major:
The example which the authors described is too limited. They should mention and discuss about the applicability of their method. For instance, does the method work well for the case of homo-multimer? How many chains can the method handle maximally and simultaneously?
To ensure the reproductivity, it is desirable to open the entire pipeline of the method (including the developed source codes and the input files of the existing softwires used in the protocol).
It is also desirable that the author show the clear (or even rough) guideline on when this method is useful or not.
Minor:
The authors may be able to introduce and discuss some more about methods, such as CR-I-TASSER and EMBuild, using deep learning.
Does the method work well for samples containing covalent ligands?
p. 6 legend of Figure 2.
((e) -> (e)
Author Response
Comments:
The authors have developed the method for generating an atomic model of a protein complex, based on the cryo-EM map. Their method resolves simultaneously a segmentation problem of the EM map and a construction problem of atomic models for all chains in a protein complex. To this end, they try to use predicted contacts between amino acids, derived from both evolutionary couplings and sequence conservation. Their algorithm proposes a segmentation of the EM map based on the predicted contacts. They show the applicability of their method for the F420-reducing hydrogenase heterotrimer and show the applicability of their modified method, which utilize the obtained backbone trace, for the wedge complex of the bacterial T6SS baseplate.
Major:
The example which the authors described is too limited. They should mention and discuss about the applicability of their method. For instance, does the method work well for the case of homo-multimer? How many chains can the method handle maximally and simultaneously?
We have added results for 2 more examples: a) a long single chain protomer in an homo-tetramer (beta-galactosidase) and a 4 chains complex (gamma-secretase). The results are presented in the new Results section “Additional challenging targets” and in the Supplementary Information.
For homo-multimers, the ASU is the monomer, so it would be a single chain. Contact prediction may contain contacts corresponding to inter-monomer interactions that cannot be considered during the fragment merging and contact map alignment procedure, possibly reducing its efficacy. We have added the following in the discussion on this point:
“For homo-multimers, the ASU corresponds to the protomer structure, so our approach would model it as a single chain. Of course, contact prediction may contain contacts corresponding to inter-protomer interactions that would not be fulfilled during our fragment merging and contact map alignment procedure, and could in principle impair its efficacy.”
To ensure the reproductivity, it is desirable to open the entire pipeline of the method (including the developed source codes and the input files of the existing softwires used in the protocol).
We have added the link to the Github repository of our algorithm (https://github.com/bougui505/EMEC) in the “Data availability” section. The repository contains the source code, usage guidelines and example data.
It is also desirable that the author show the clear (or even rough) guideline on when this method is useful or not.
We believe that the enhanced presentation of the limitations of our approach in the Discussion and the results for the two additional examples outline sufficiently the range of applicability and usefulness of our method.
Minor:
The authors may be able to introduce and discuss some more about methods, such as CR-I-TASSER and EMBuild, using deep learning.
We have added references to the 2 recent deep-learning based tracing tools mentioned by the reviewer, namely EMBuild (doi: 10.1038/s41467-022-31748-9) and CR-I-TASSER (doi: 10.1038/s41592-021-01389-9) and reformulated this section of the introduction:
“Recent state-of-the-art approaches include DeepTracer [13], EMBuild [14] or CR-I-TASSER [15] that predict main-chain probability maps for the positioning of Cα atoms in the density. These DL methods are usually trained on large sets of experimental and simulated EM maps along with the corresponding known PDB models”
Since the main focus of the manuscript is not on a deep-learning approach, we feel it is not relevant to discuss more about such methods.
Does the method work well for samples containing covalent ligands?
When ligands are present and covalently bound to the protein chain, such as glycosylation for instance, the corresponding density would in principle remain empty during fragment tracing. If some calpha beads would be positioned in the ligand-density, it would be either pruned during graph generation since the method does not generate branched fragments or fill with a separate fragment that would not be later merged in the main chain since no contacts would be found.
- 6 legend of Figure 2.
((e) -> (e)
Corrected.
Reviewer 3 Report
This manuscript describes a graph-based method for de novo model building into cryo-EM maps with medium resolution. Residue co-evolution is also taken into consideration, which could improve the resulting model. For readers in the cryo-EM field, this is a method worth learning. Therefore, I would recommend the publication of this paper if the following concerns can be reasonably addressed.
1. Can the authors explain more on how to use their algorithm? Maybe a brief step-by-step manual as a supplementary material? The authors should also consider share their source code for the public to use and give an instruction on how to prepare the inputs and how to interpret the outputs.
2. The last paragraph on page 7 (lines 223-229) describes how the model was compared with the predicted contact maps. However, the authors did not describe any findings or results from this comparison. From Figure 4, it looks like majority of the structure is in red, meaning low overlap score. Does this mean that the model is not accurate at all, or is there any limitation by only comparing the overlap score? The authors should discuss these in the paper. Also, the title of Figure 4 should be re-written to give a more concise summary of what is shown in the figure.
3. Can the authors show more examples of using this mothed for model building? How general is it? Any advantages and limitations? Is there a size limitation (number of subunits, molecule mass, homo vs. hetero complexes) on the complex that can be built using this method?
4. Can the authors elaborate more on the “crossed” features in chain A and C in Figure 3? Is the method smart enough to reverse these regions on its own, or manual intervention is needed?
5. A few minor points: Line 30, “Below 3.3” can be changed to “Better than 3.3”; line 36, “Above 4.5” can be changed to “Worse than 4.5”; line 326, “form” should be “from”.
Author Response
Comments:
This manuscript describes a graph-based method for de novo model building into cryo-EM maps with medium resolution. Residue co-evolution is also taken into consideration, which could improve the resulting model. For readers in the cryo-EM field, this is a method worth learning. Therefore, I would recommend the publication of this paper if the following concerns can be reasonably addressed.
- Can the authors explain more on how to use their algorithm? Maybe a brief step-by-step manual as a supplementary material? The authors should also consider share their source code for the public to use and give an instruction on how to prepare the inputs and how to interpret the outputs.
We have added the link to the Github repository of our algorithm (https://github.com/bougui505/EMEC) in the “Data availability” section. The repository contains the source code, usage guidelines and example data.
- The last paragraph on page 7 (lines 223-229) describes how the model was compared with the predicted contact maps. However, the authors did not describe any findings or results from this comparison. From Figure 4, it looks like majority of the structure is in red, meaning low overlap score. Does this mean that the model is not accurate at all, or is there any limitation by only comparing the overlap score? The authors should discuss these in the paper. Also, the title of Figure 4 should be re-written to give a more concise summary of what is shown in the figure.
The contacts of the final model are accurate except for the regions where the predicted contacts are sparse as already stated in the manuscript text. When comparing with the contacts of the reference PDB model, the overlap scores are much higher, indicating that the contacts in the model are accurate. The comparison with the predicted contacts was intended to represent a real case situation, when the ground-truth model is not known. We have now added a figure in the supplementary information (Figure S1) showing the contact overlap scores when comparing the reference PDB structure and the reconstructed model, and added the following in the Results section:
“Even if the average overlap score with the predicted contacts is rather low, it still allows to identify regions with poor overlap and potentially incorrect sequence assignment. The overlap scores are much higher when comparing with the contacts observed in the true solution, i.e. the reference PDB structure, but such a comparison is possible in a real-case scenario when the true solution is not known (Figure S1).”
The caption of Figure 4 was changed with a more explicit title: “Overlap between predicted contact maps and contact map of the final model”
- Can the authors show more examples of using this mothed for model building? How general is it? Any advantages and limitations? Is there a size limitation (number of subunits, molecule mass, homo vs. hetero complexes) on the complex that can be built using this method?
We have added results for 2 more examples: a) a long single chain protomer in an homo-tetramer (beta-galactosidase) and a 4 chains complex (gamma-secretase). The results are presented in the new Results section “Additional challenging targets” and in the Supplementary Information.
For homo-multimers, the ASU is the monomer, so it would be a single chain. Contact prediction may contain contacts corresponding to inter-monomer interactions that cannot be considered during the fragment merging and contact map alignment procedure, possibly reducing its efficacy. We have added the following in the discussion on this point:
“For homo-multimers, the ASU corresponds to the protomer structure, so our approach would model it as a single chain. Of course, contact prediction may contain contacts corresponding to inter-protomer interactions that would not be fulfilled during our fragment merging and contact map alignment procedure, and could in principle impair its efficacy.”
Number of subunits and chain length (i.e. number of residues) might also be other possible limitations since i) the merging procedure relies on a full pairwise estimation of the fragments pairing and for which the complexity increases along with the number of fragments and ii) each chain is processed sequentially. We have added the following paragraph toward the end of the discussion
“Number of subunits and chain length (i.e. number of residues) might also limit the efficacy of our approach since i) the merging procedure relies on a full pairwise estimation of the fragments pairing and for which the complexity increases along with the number of fragments and ii) each chain is processed sequentially. As demonstrated with the two additional examples, missing fragments for low density regions also hampers the correct registering of the sequence with the predicted contacts.”
- Can the authors elaborate more on the “crossed” features in chain A and C in Figure 3? Is the method smart enough to reverse these regions on its own, or manual intervention is needed?
No manual intervention is needed to find the orientation of the fragments. The method tests both directions during the merging procedure as already stated in the manuscript . By definition, the generated fragments have no directionality. The reverse fragments presented as “crossed-features” in Figure 3 arise from the given initial numbering of Calpha beads in the fragments for comparison purposes. What the figure highlights is that the Calpha ordering is correct but the initial numbering was fully reversed and later corrected, automatically, during fragment assembly/merging.
- A few minor points: Line 30, “Below 3.3” can be changed to “Better than 3.3”; line 36, “Above 4.5” can be changed to “Worse than 4.5”; line 326, “form” should be “from”.
Corrected.
Round 2
Reviewer 1 Report
The corrections by the authors address my concerns and I support publication.
Reviewer 2 Report
In line 323,
"The gama-secretase" should be "The gamma-secretase"?
In line 330,
"... to chain B (Figure S2A)." should be "... to chain B (Figure S2B)." ?